# FIND Stroke Recovery Study (FIND): rationale and protocol for a longitudinal observational cohort study of trajectories of recovery and biomarkers poststroke

Cecilia Brännmark [1,2] Sofia Klasson,[1] Tara M Stanne [1,3]
Hans Samuelsson [4,5] Margit Alt Murphy [6,7] Katharina S Sunnerhagen [7]
N. David Åberg,[8,9] Oscar Jalnefjord [10,11] Isabella Björkman-Burtscher [12,13]
Katarina Jood,[7,14] Turgut Tatlisumak [7,14] Christina Jern[1,3]

KJ, TT and CJ contributed equally.

For numbered affiliations see end of article.

**Correspondence to**
Dr Cecilia Brännmark;
cecilia.brannmark@vgregion.se

## ABSTRACT

**Introduction** Comprehensive studies mapping domain-specific trajectories of recovery after stroke and biomarkers reflecting these processes are scarce. We, therefore, initiated an exploratory prospective observational study of stroke cases with repeated evaluation, the *FIND Stroke Recovery Study*. We aim to capture trajectories of recovery from different impairments, including cognition, in combination with broad profiling of blood and imaging biomarkers of the recovery.

**Methods and analysis** We recruit individuals with first-ever stroke at the stroke unit at the Sahlgrenska University Hospital, Sweden, to FIND. The inclusion started early 2018 and we aim to enrol minimum 500 patients. Neurological and cognitive impairments across multiple domains are assessed using validated clinical assessment methods, advanced neuroimaging is performed and blood samples for biomarker measuring (protein, RNA and DNA) at inclusion and follow-up visits at 3 months, 6 months, 1 year, 2 years and 5 years poststroke. At baseline and at each follow-up visit, we also register clinical variables known to influence outcomes such as prestroke functioning, stroke severity, acute interventions, rehabilitation, other treatments, socioeconomic status, infections (including COVID-19) and other comorbidities. Recurrent stroke and other major vascular events are identified continuously in national registers.

**Ethics and dissemination** FIND composes a unique stroke cohort with detailed phenotyping, repetitive assessments of outcomes across multiple neurological and cognitive domains and patient-reported outcomes as well as blood and imaging biomarker profiling. Ethical approval for the FIND study has been obtained from the Regional Ethics Review Board in Gothenburg and the Swedish Ethics Review Board. The results of this exploratory study will provide novel data on the time course of recovery and biomarkers after stroke. The description of this protocol will inform the stroke research community of our ongoing study and facilitate comparisons with other data sets.

## STRENGTHS AND LIMITATIONS OF THIS STUDY

⇒ Time-resolved data of poststroke recovery in a sample of consecutively recruited and well-characterised stroke patients of all ages.
⇒ Follow-up includes detailed outcome metrics in multiple domains, including motor, cognitive and functional outcome after stroke.
⇒ Broad, repetitive biomarker profiling, in blood and using advanced brain imaging.
⇒ Heterogeneity in the time point of sample collection in the acute phase.
⇒ Persons suffering the most severe strokes are likely to be under-represented in the data, skewing results towards mild strokes.

**Trial registration number** The protocol is registered at http://www.clinicaltrials.gov, Study ID: NCT05708807.

## INTRODUCTION AND RATIONALE

Stroke is the most common cause of adult disability,[1] and the number of stroke survivors with disabilities is increasing.[2] Common stroke survivor disabilities include motor, cognitive and visual impairments as well as depression, limitations in activities of daily living, restrictions in participation and decreased quality of life. One of the first questions asked by newly diagnosed stroke patients' and their next of kin often concerns recovery and prognosis. Today, that is not possible to adequately answer as the knowledge on timelines, mechanisms and individual variations as well as predictors of recovery are not sufficient. On the group level, type of stroke, stroke size and location of brain lesion, acute treatment, rehabilitation, depression, type 2 diabetes, hypertension, other comorbidities and living alone[3,4] are all factors

that have been found to influence recovery after stroke. However, today knowledge on how to predict outcome and recovery after stroke on the individual level is limited as some of the variations cannot be explained by clinical parameters.[5] Also, the full range of mechanisms and factors leading to recovery are not precisely known, and the molecular pathways that are involved in brain injury and repair in stroke are likely to vary over time.[6] For example, collateral circulation, reperfusion and haemorrhagic transformation have key roles in the acute stage, while the determinants of neuroplasticity become important later.[7]

The rate of recovery also differs substantially between type of impairment and stroke survivors[7] and can be influenced by comorbidities and lifestyle. A majority of the existing studies conclude that most recovery occurs during the first 3–4 months after index stroke.[8–11] However, it is of note that these studies have focused on specific motor functions and global measurements of disability (eg, modified Rankin Scale (mRS)) and do not capture the whole complexity of recovery after stroke and that there are data-supporting domain-specific recovery patterns. For example, impairment in cognitive function is common during the early stage after stroke (in at least 75%),[12–14] and successive improvement is described during the first 6 months to a year.[7 9] A recent review on recovery reported the largest improvement in cognitive function 61–180 days (or 2–6 months) after stroke onset.[15] Knowledge about the long-term trajectory after 1 year is more uncertain, some report stability in cognitive function between 3 months and 2 years after stroke for the majority of cases,[16] while others found a decline at the group level.[17] There are also studies reporting that while some stroke survivors display no change in cognition or a decline, there are also stroke survivors who have improved cognitive function at long-term follow-ups.[18–20] Data suggest that for some cognitive impairments, such as aphasia and neglect, recovery may take place even years after index stroke.[21]

Further studies of the long-term trajectories with sensitive measures of both global and specific domains of cognition and of possible mechanisms and modulators behind the change are warranted. In addition to the FIND study presented here, there are some additional ongoing studies on this topic such as OX-CHRONIC, STROKE-Cog, DISCOVERY, ODYSSEY, R4VaD and APPLE,[22–27] all with different focuses regarding age groups and research questions. Increased knowledge in this field is important since it can be used to answer the questions on prognosis raised by patients and kin. Also, cognitive deficits after stroke are a major contributor to the burden of stroke and loss of quality of life. Moreover, we lack specific adjuvant treatments to enhance recovery after stroke. To establish such treatments increased knowledge on the pathways involved in stroke recovery and the trajectory of their activity is necessary, as has previously been suggested.[28]

Developing blood biomarkers to monitor recovery trajectories and to assess when a certain treatment might benefit a certain patient would increase possibilities for developing new rehabilitation interventions after stroke and allow more individualised and precise treatment planning.[29] This knowledge can only be obtained by collecting a broad battery of clinical measures in combination with molecular profiling at set timepoints ranging from the acute phase to the chronic stages.[7]

Here, we aim to investigate the rate, pattern and degree of recovery after stroke in various domains. We use validated clinical measures, including broad assessment of cognitive functioning as well as repeated neuroimaging and blood draws for analyses, in the acute phase and at 3, 6 and 12 months as well as 2 and 5 years after stroke onset. In addition, we classify the aetiological subtypes of all strokes.

## METHODS AND ANALYSIS
### Inclusion and exclusion criteria
We enrol patients presenting with first-ever ischaemic stroke or intracerebral haemorrhage admitted to the stroke units at the Sahlgrenska University Hospital in Gothenburg, Sweden. Recruitment started January 2018 and involved written informed consent as described below. We plan to include 500 stroke cases or more, and we estimate to enrol until 2026 with follow-ups ongoing until all participants have had their 5-year follow-up (ie, 2030). During the COVID-19 pandemic, study participants had to be followed up remotely. We, therefore, aim to have at least 300 participants who have been able to come to ≥3 in person follow-up visits. We will keep track of attrition with regards to participation rates in study visits and adjust the time line for inclusion accordingly. By the end of February 2023, 305 stroke cases had been enrolled and 5 cases had been at their last 5-year follow-up visit.

The inclusion criteria are:
1. ≥18 years of age.
2. First-ever acute ischaemic stroke or intracerebral haemorrhage.

The exclusion criteria are:
1. Prestroke mRS score of ≥3.
2. Severe neurodegenerative disease, cerebral neoplasm or terminal illness and patients considered unlikely to be able to participate in study procedures during follow-up visits. However, we invite study participants with severe cognitive impairments and/or communication difficulties because these persons can often participate fully or in parts of the study procedures with assistance of a next-of-kin, friend, or caregiver.

### Patient and public involvement
Patients with living experience of stroke were involved in FIND during the design of the study. A survey with stroke survivors and representatives from stroke patient association was used to identify which questions and areas that were most important. Better prognostic information and cognitive deficits emerged as important areas. We, hence, designed our study to address those topics. Initially, our

ambition was to include a large number of metrics in order to comprehensively capture different outcomes (see online supplemental table 1 for original protocol). However, we found that such an extensive protocol left us with substantial missing data. After a new survey taking into account the respondent burden, we chose to omit some cognitive tests and a few patient-reported outcome measures (PROMs) capturing overlapping outcomes in the final protocol.

Study inclusion started in January 2018, however, due to COVID-19-related restrictions, we had to modify the study protocol and pivot to remote testing via telephone during parts of 2020–2021, in line with other studies.[22] We then used a shorter version of the Montreal Cognitive Assessment (MoCA) which has been validated for this setting.[30–32]

### Assessments at baseline during the hospital stay at the stroke unit

The neurologic deficits at admission are assessed using the National Institutes of Health Stroke Scale (NIHSS; total score and scores for all subscales). Prestroke disability is assessed by prestroke mRS. We also assess perceived health status 4 weeks prestroke using the Stroke Impact Scale (SIS), and here we use the subscales for strength, memory and thinking, communication, mobility and participation.

Patients are characterised regarding vascular risk factors, personal medical history including comorbidities, social situation, education, employment and lifestyle factors such as smoking and physical activity. A first-degree family history of stroke and transient ischaemic attacks and other cardiovascular diseases is obtained using a written questionnaire and structured interview. If a participant is unable to provide an adequate family history, a collateral history from a relative is sought. Weight, height and blood pressure are measured. We register acute interventions, rehabilitation, other treatments and complications such as infections (including COVID-19) during hospital stay. For patients treated with recanalisation, that is, thrombectomy and/or thrombolysis, we register the NIHSS score after this treatment.

In addition to the NIHSS motor arm subscore, motor impairment of the upper extremity is assessed by trained physiotherapists (instructed by author MAM) using the Fugl-Meyer assessment of upper extremity (FMA-UE) and shoulder abduction and finger extension (SAFE) score. Walking ability is assessed by functional Ambulation Category[33] and 10 m walking test and postural control by the Berg Balance Scale (BBS).[34] Global cognitive function is assessed by MoCA.[35] In addition, we use the following specific subtasks from MoCA: (1) attention/working memory (target detection, serial sevens and digit forward and backward); (2) executive function (trails B task, phonemic fluency and verbal abstraction); (3) visuospatial construction (three-dimensional figure copy and clock drawing); (4) episodic memory (five-word recall) and (5) language (naming task and sentence repetition).

Participants with aphasia also perform the 15-item Boston Naming test (BNT) and items of language comprehension from Addenbrooke's Cognitive Examination-Revised test (ACE-R) and from 'Norsk grunntest for afasi' (NGTA). Participants are encouraged to complete as much of the tests as possible, and if completion is not possible, the cause is noted to allow for analysis of possible confounders such as aphasia giving false low scores in cognitive tests of other domains.

### Neuroimaging

During the hospital stay at the stroke unit all patients undergo computerized tomography (CT) of the brain and if indicated magnetic resonance imaging (MRI) as part of the clinical routine. In case the MRI is not routinely included, all patients will be offered a brain MRI examination according to a study-specific protocol. The protocol includes morphological, functional and microstructural imaging rendering structural, diffusion and perfusion metrics. From brain MRI, we will extract different imaging biomarkers such as lesion location and size, tissue characteristics such as diffusion measures and prestroke cerebral tissue condition including atrophy and white matter condition.[36]

### Assessments at follow-up visits

Surviving participants are invited to follow-up visits at the hospital at 3 and 6 months and 1, 2 and 5 years after the index stroke (figure 1 and cognitive tests are detailed in table 1). At all visits, functional outcome is scored using the mRS, the degree of neurological impairments is assessed using NIHSS (total score and scores for all subscales), and cognitive outcome is evaluated using the same tests as described above for the acute stage. In addition, tests of attention, neglect, processing speed, executive functions, basic perception and motor speed are used such as the Trailmaking Test (TMT) A, B[37] and D and Random Shape Cancellation test[38] (RSCT) with and without 30 s time restriction (version A and B). For study participants under 55 years of age at baseline, we include extended testing of processing speed, selective attention, memory and executive functions such as inhibition and verbal fluency using part 1–3 of Color-Word Interference test (CWT) in Delis-Kaplan Executive Function System,[39] the 10-word test from Repeatable Battery for the Assessment of Neuropsychological Status[40] and the Verbal Fluency Test (FAS).[41] Participants with aphasia also perform the 15-item BNT[42] and items of language comprehension from ACE-R and from NGTA.

For participants with motor deficits at inclusion, follow-ups with FMA-UE, SAFE, functional Ambulation Category, 10 m walking test and BBS are included. For all participants, we gather PROMs, including the selected subscales of SIS[43] described above as well as the SIS visual analogue scale on overall recovery, the Daily Fatigue Impact Scale (D-FIS),[44] a life satisfaction question on contentment with life in general from the Life Satisfactory Questionary[45] and the Hospital Anxiety and Depression

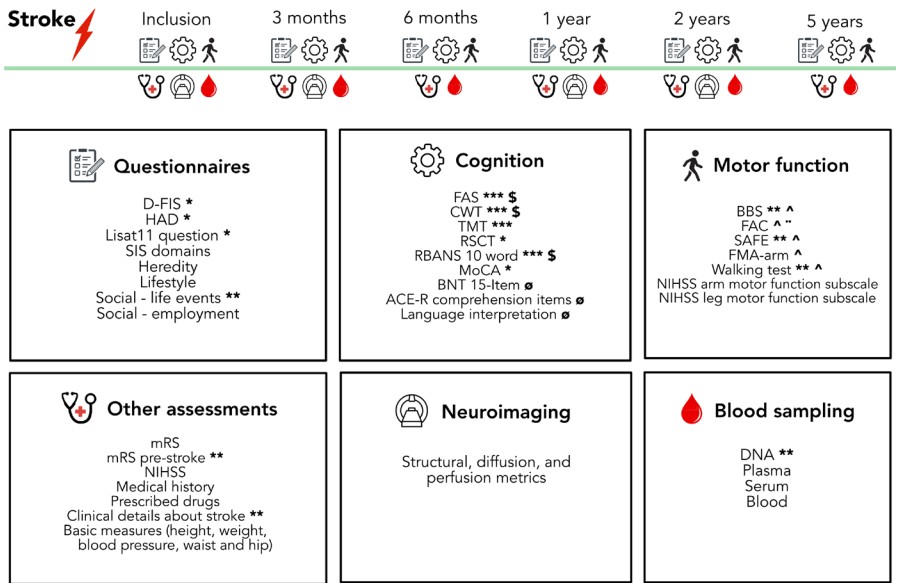

**Figure 1** Overview of study protocol. *Performed at all visits except for inclusion. **Only at inclusion. ***Not conducted at inclusion or at 6 months follow-up to prevent repetition bias. ·· Performed at all visits except at 1 year. $ for participants <55 years old at index stroke. ø If aphasia at inclusion. ^ If deficits at inclusion. These fixed time points are complimented by register studies that continously monitor recurrent stroke and other vascular events, medication, and deaths. ACE-R, Addenbrooke's Cognitive Examination-Revised test; BNT, the 15-Item Boston Naming Test; BBS, Berg Balance Scale; CWT, Color-Word Interference test; D-FIS, Daily Fatigue Impact Scale; FAC, functional Ambulation Category; FMA, Fugl-Meyer Assessment of Upper Extremity; FAS, Verbal Fluency Test; HADS, Hospital Anxiety and Depression Scale; Lisat11, Life Satisfactory Questionary; mRS, Modified Rankin Scale; MoCA, Montreal Cognitive Assessment; NIHSS, National Institutes of Health Stroke Scale; RSCT, Random Shape Cancellation Test; RBANS, Repeatable Battery for the Assessment of Neuropsychological Status; SAFE, Shoulder Abduction and Finger Extension; SIS, Stroke Impact Scale; TMT, Trailmaking Test.

Scale (HADS).[46] At each follow-up visit, we also register clinical variables known to influence outcomes and/or changes in these variables since last visit, such as rehabilitation attained, other treatments, vascular risk factors, physical activity, socioeconomic status, infections (including COVID-19) and other comorbidities. Moreover, all study participants are invited to repeated neuroimaging with a study-specific protocol at 3–6 months as well as 1 and 2 years after stroke onset.

To ensure that individuals with the worse outcome/socioeconomic situations are not lost to follow-up, we collect the data in the following priority order: (1) patients receive questionnaires by mail and come to the hospital for tests, neuroimaging and blood draw; (2) patients receive mail questionnaires and a research nurse conducts a telephone interview; (3) patients receive mail questionnaires only; (4) when none of the above can be implemented, patients will be followed through national registers including Riksstroke,[47] provided they have not chosen to withdraw their participation.

### Data retrieval
Data on deaths, and recurrent stroke or other major vascular events, comorbidities and prescribed drugs over time during the whole study period are obtained from the Swedish Population Register, the National Patient Register, the Prescribed Drugs Register and the National Register on Cause of Death as described.[48] For events in

the Patient Register and the Cause of Death Register, we review medical records.

For patients who are unable to participate in our 3-month and/or 1-year follow-up, data on outcomes are obtained from Riksstroke,[47] which has data on self-reported outcomes at 3 and 12 months after stroke. The Riksstroke data are transformed to the modified Rankin Scale (mRS) scores 0–2, 3, 4 or 5 using a validated algorithm as described.[49] Data on health and social support are obtained from the National Board of Health and Welfare and data on education and employment status from the Longitudinal Database of Education, Income and Occupation Register (Swedish acronym: LISA for *Longitudinell integrationsdatabas för sjukförsäkrings- och arbetsmarknadsstudie*). Data on management of diabetes and hypertension are obtained from the National Diabetes Register and the Primary Care Registry, respectively.

### Stroke subtyping
Ischaemic stroke subtyping is performed by a trained physician using validated systems (Trial of Org. 10 172) in Acute Stroke Treatment criteria,[50] the Causative Classification of Stroke system[51] and the Oxfordshire Community Stroke Project (OSCP) classification[52] for ischaemic stroke. Intracerebral bleedings (ICH) are classified using SMASH-U (Structural lesion, Medication, Amyloid angiopathy, Systemic/other disease, Hypertension, Undetermined).[53]

**Table 1** Cognitive tests in the *FIND* study

| Time points | Tested cognitive domain | Tests |
|---|---|---|
| At all assessments | Global cognitive function | MoCA total score |
| Specific cognitive domains | | |
| At all assessments | Executive functions | MoCA—short trails B task, phonemic fluency and verbal abstraction |
| | Attention/working memory | MoCA—target detection, serial sevens and digit forward and backward; |
| | Visuospatial construction | MoCA—three-dimensional figure copy and clock drawing |
| | Episodic memory | MoCA—5-word recall |
| | Language | MoCA—naming task and sentence repetition |
| At all assessments, for patients with aphasia at acute stage | Language—Aphasia | 15-Item Boston Naming Test; language comprehension from Addenbrooke's Cognitive Examination-Revised test and from Norsk grunntest for afasi |
| At all follow-ups | Basic perception and motor speed | Trail making test-D (connecting empty circles by following a predetermined dotted track as fast as possible) |
| | Processing speed | Random Shape Cancellation Test with time restriction (RSCT-version B) |
| | Visuospatial attention | Total score of Random Shape Cancellation Test without time restriction (RSCT-version A) |
| | Attention— Neglect | Asymmetry score of Random Shape Cancellation Test without time restriction (RSCT-version A) |
| At follow-up 3 months, 1 year, 2 years and 5 years | Processing speed | Trail making test A |
| | Executive functions | Trail making test B |
| Extended testing participants <55 years | | |
| At follow-up 3 months, 1 year, 2 years and 5 years | Processing speed | Part 1 and 2 from Color-Word Interference test (CWT) in Delis–Kaplan Executive Function System (D-KEFS) |
| | Memory | 10-word test from Repeatable Battery for the Assessment of Neuropsychological Status (RBANS) |
| | Executive functions—selective attention and inhibition | The Inhibition part in Color-Word Interference test in D-KEFS |
| | Executive functions—fluency | The Verbal Fluency test (FAS) |

MoCA, Montreal Cognitive Assessment.

## Blood sampling

A first blood sample is drawn as soon as possible after admission to the stroke unit, and on a subgroup, we also draw a sample 7 days after stroke. Thereafter, blood samples are drawn at each follow-up visit. On all occasions, venous blood is collected after an overnight fast. Blood is drawn in several tubes and handled at room temperature, whole blood or supernatants, that is, plasma and serum, are robot aliquoted within 2 hours and stored in freezers at Biobank Väst, a part of the Swedish National Biobank, pending analysis, for details, see table 2. Blood is also drawn for analyses of total leucocyte count and differential blood count at the Dept of Clinical Chemistry at our hospital.

## Blood biomarker analysis

Clinically established blood biomarkers are commonly based on measurements of protein concentrations in plasma or serum. We and others have, for example, previously found plasma proteins with subtype-specific patterns,[54 55] protein concentrations associated with recurrent vascular events[56 57] as well as neurological and functional outcomes.[57–60] Here, planned analyses include multiomics profiling, on the metabolite, protein, RNA and DNA level, utilising the Swedish National infrastructure for state-of-the-art omics at Science for Life Laboratory (SciLifeLab), Uppsala and Stockholm, and the Bioinformatics and Data Centre, Gothenburg.

## STUDY AIMS
### Primary objective

The primary aims are twofold:
1. Determine trajectories of global as well as specific domains of cognition and neurological function after stroke.

**Table 2** Blood sample handling

| Type | Tube(s) and additive | Centrifugation | Storage |
|---|---|---|---|
| Whole blood | With 10%* ethylenediaminetetraacetic acid (EDTA)† | NA | −80°C |
| EDTA plasma | With 10%* $^{\alpha}$ EDTA† | 2000 g for 10 min | −80°C |
| Citrate plasma | With 3.2%* natrium citrate† | 2500 g for 20 min | −80°C |
| Serum | Serum separator tube† | Left to coagulate 30–60 min and spun 2000 g for 10 min | −80°C |
| Whole blood | Tempus Blood RNA‡ | NA | −20°C |

*Volume/volume.
†Becton, Dickinson and Company, Becton Drive Franklin Lakes, NJ, USA.
‡Life Technologies, Carlsbad, CA, USA.
NA, not applicable.

2. Identify clinical variables as well as neuroimaging and blood biomarkers, predicting poststroke cognitive and neurological outcomes.

## Secondary objectives

1. Determine trajectories of blood biomarker concentrations after stroke and optimal time points for measurements.
2. Determine trajectories of upper arm motor function after stroke.
3. Determine trajectories of self-reported depressive symtoms, fatigue, participation and quality of life after stroke, and their relations to other outcome measures with focus on cognition.
4. Identify stroke subtype-specific recovery trajectories.
5. Identify clinical variables, as well as neuroimaging and blood biomarkers, predicting upper arm motor function and self-reported outcomes after stroke.

## Organisation

The study is conducted with three principal investigators CJ, TT and KJ. The patients are recruited at the stroke units at the Sahlgrenska University Hospital in Gothenburg. All follow-up invitations and visits are managed by specially trained research nurses. Data analysis will be overseen by the CJ and KJ research groups.

## ETHICS AND DISSEMINATION
### Ethical approval

This study was approved by the Regional Ethics Review Board in Gothenburg, and the Swedish Ethical Review Authority (REC number 187–17 and 2021-06071-02, respectively, Guarantor: CJ). Written informed consent is obtained from all participants prior to enrolment. For participants who cannot give written consent, consultant consent is obtained from their next-of-kin. Participants may withdraw their consent at any time. If so they can choose whether data collected prior to withdrawal can be used or deleted and counted as missing.

## Dissemination

The protocol is registered at http://www.clinicaltrials.gov.

## DISCUSSION

As stated in the Introduction section, there is surprisingly little data on trajectories (ie, repeated measures) of stroke recovery in several domains in itself and even less in combination with biomarkers. This protocol describes the *FIND Stroke Recovery Study*, a longitudinal cohort study in stroke with detailed phenotyping and repetitive assessments of outcome measures across multiple domains coupled to biomarker identification. We perform broad cognitive testing and use NIHSS and mRS, the most used measures of neurological deficits and functional outcome, respectively, and several motor function measures, essentially in accordance with recently published recommendations.[61–63] We also use several complementary outcome metrics, objectively tested as well as self-reported, to comprehensively capture a broad spectrum of outcomes including different dimensions according to the International Classification of Functioning, Disability and Health. In combination, we collect blood samples for broad biomarker profiling as well as stroke lesion data using advanced neuroimaging. The field of predictive biomarkers in ischaemic stroke has not yet reached a conclusive state and even between high-quality study results are to some extent inconsistent.[59] A few ongoing studies have designs similar to FIND and include measurements of neuroimaging and/or blood biomarkers: for example, STROKE-Cog, R4VaD. OX-CHRONIC, APPLE and DISCOVERY[22–24 26 27] including adult stroke survivors. The protocols of these studies have follow-ups allowing for comparisons with some of the cognitive outcomes in our study protocol making external validation of some of our findings possible. However, in comparison to these, FIND is more focused on blood-based protein and RNA biomarkers and notably includes blood sampling at each study visit. In line with our protocol, DISCOVERY[24] also includes MRI and blood-based protein biomarkers,

whereas STROKE-Cog has a focus on circulating blood biomarkers of immunological responses.[23] R4VaD[26] is mainly focused on dementia after stroke and includes blood biomarkers, whereas OX-CHRONIC[22] includes neuroimaging, but not biomarkers.

Cognitive outcome after stroke is of great importance for the patients, their relatives and care givers. Despite this, trajectories of cognitive recovery after stroke and mechanisms behind the recovery are less studied than motor and other functionalities,[64] likely due to the complexity of cognition. Thus, we chose to add several tests assessing similar domains to capture different aspects and nuances of cognition. We use the commonly used MoCA and add tests of different specific cognitive domains such as processing speed (eg, TMT-A and CWT 1–2), attention (eg, RSCT), executive functions (eg, TMT-B and CWT 3), memory (10-word test from repeatable battery) and language and verbal fluency (BNT and FAS). These tests capture specific cognitive domains as well as domains with potentials for a more global influence on cognitive function such as executive functions, complex attention and information processing speed. The latter tests are included because we believe that these are understudied in stroke. Moreover, such tests might be more sensitive to monitoring impairments than domain-specific measures in long-term follow-ups after stroke. We include additional cognitive tests for participants <55 years at inclusion than for those >55 years. The rationale for a more detailed cognitive mapping in younger participants is their longer expected life-time meaning many years of productive life living with impairments in combination with the generally higher expectations and demands on cognitive function. Moreover, we know from experience that the ability of stroke survivors to perform a large battery of cognitive tests concentrated to 1 day is better in the young than in the older population. There is another ongoing prospective study of trajectories of cognitive function after stroke, ODYSSEY[25] focusing on stroke in young adults. However, this study is not focused on cognition and does not include blood biomarkers. To complement the objective cognitive tests, we also include self-reported cognitive function (eg, SIS). Simultaneous to assessments of cognitive outcome, we study neurological outcomes with NIHSS total and subscale scores as well as finer-grained assessments of upper arm motor function. We also assess functional outcome by the commonly used mRS and include PROMs (eg, domains of SIS) to complement our objective functionality measures. Moreover, as mRS is only moderately correlated to quality of life,[65] we include a question on life satisfaction.

Stroke is associated with increased rates of depression, and a previous study has found about one-third of stroke patients to be affected by poststroke depression.[66] A correlation between depressive symptoms (HADS-D>7) and performance in cognitive tests both on the global and domain-specific level has been reported.[67] Also, depressive symptoms measured as HAD score >4 have been found to correlate to self-reported memory and thinking problems (using SIS).[68] Because depression and cognitive impairment seem to be connected and reciprocally influential in stroke survivors, we are assessing mental health trajectories in our cohort. To enable comparison of our results with those from as many studies as possible, we decided to include the widely used HADS which also includes anxiety.

Fatigue, a common symptom after stroke, can also add to cognitive dysfunction. Therefore, we included D-FIS to measure the impact of fatigue on cognitive recovery in this cohort. D-FIS is highly correlated to the more extensive Fatigue Impact Scale[69] but less time-consuming for the patient.

Aetiologic subtype of ischaemic stroke has been reported to associate not only with cognitive outcome[15] but also with other aspects of poststroke outcome such as persisting neurological deficits.[70] The study protocol presented here allows for subtyping of all ischaemic strokes according to two validated systems and subtype-specific analysis of outcome measures. In line with this, we also have the possibility to classify the haemorrhagic stroke's aetiology into structural vascular lesions, medication, amyloid angiopathy, systemic disease, hypertension or undetermined using the SMASH-U[53] classification system and to correlate the subtypes to biomarkers and outcome measures.

Acute imaging in stroke using CT and/or MRI aims primarily at identification of location and size of a lesion, exclusion of haemorrhage prior to treatment, differentiation between irreversibly affected brain tissue and tissue at risk as well as identification of stenosis or occlusion in supplying vessels. We also perform neuroimaging with MRI at inclusion and perform detailed mapping of location and size of the lesion. Imaging can, however, also provide insights into prestroke status of the brain as well as poststroke tissue integrity within the lesion and in normal appearing cerebral tissue. Pre-existing disease markers, such as white matter lesions, atrophy, microinfarcts and comorbidity of other aetiology, and poststroke tissue integrity markers, such as metabolic, diffusion and perfusion metrics, aid us to monitor disease progression and to target outcome prediction.[36] We also perform MRI at three time points poststroke in order to assess for instance white matter condition.

### Strengths and limitations

The strengths of the present study include time-resolved data of poststroke recovery in a sample of consecutively recruited and well-characterised stroke patients of all ages. Moreover, the follow-up includes a comprehensive set of detailed outcome metrics with focus on cognitive and neurological function with global as well as a domain-specific measures. Taken together with our broad, repetitive biomarker profiling, in blood and using advanced MRI, this allows for exploration of biomarkers for cognitive, neurological and functional outcomes as well as fatigue and depressive symptoms after stroke. Notably, this study is in accordance with guidelines for stroke outcome

studies, which has been published after initiation of the present protocol.[61] Further strengths lie in the design of our protocol, which was influenced by a referendum with study participants, research nurses and representatives from the local stroke patient association. Nonetheless, our study has important limitations. A limitation is heterogeneity in the time point of sample collection in the acute phase. For practical reasons, it is not always possible to include participants withing 24 hours after symptom onset. We prioritise to be able to include as many study participants as possible and document the time of sampling to enable analysis of impact of the sampling time on the data. Another limitation, shared with most long-term follow-up studies after strokes, is that persons suffering the most severe strokes will be under-represented in the data, skewing results towards mild strokes. This is due to both lower inclusion rate and higher likelihood for withdrawal from the study followups in severe stroke patients. To our advantage, we opt for telephone follow-ups when a physical visit is not possible. Moreover, very few study participants are fully lost as even though they will not come to follow-ups, we can still follow them through national registers, such as the Riksstroke Register, and the National Patient Register with consent from the participant. Also, due to the COVID-19 pandemic, there will be a period of several months with lower inclusion rate and modified follow-up visits. A third limitation is that several cognitive tests used by us (and others) rely on language and the cognitive function measures of partcipants with aphasia may be affected. We have in line with the previously mentioned R4VaD[26] study decided to include participants with aphasia and encourage them to complete the study protocol, if possible, while other ongoing studies have decided to exclude patients with aphasia.[23]

## SIGNIFICANCE/CONCLUSION

The *FIND Stroke Recovery Study* will provide an unprecedented opportunity to inform on the time course of recovery coupled to detailed biomarker analyses (imaging, multiomic profiling of protein, RNA and DNA) after stroke. The gained knowledge from this project will be helpful for designing future studies on biomarkers and interventions targeting stroke recovery. Moreover, this study will provide an opportunity to find parts of the puzzle of extent and rate of recovery for an individual. Hopefully, together with other studies in the field, it will lead to better information on prognosis to patients and their loved ones.

**Author affiliations**
[1]Department of Laboratory Medicine, Institute of Biomedicine, The Sahlgrenska Academy, University of Gothenburg, Gothenburg, Sweden
[2]Region Västra Götaland, Sahlgrenska University Hospital, Department of Research, Development, Education and Innovation, Gothenburg, Sweden
[3]Region Västra Götaland, Sahlgrenska University Hospital, Department of Clinical Genetics and Genomics, Gothenburg, Sweden
[4]Institute of Psychology, Faculty of Social Sciences, University of Gothenburg, Gothenburg, Sweden
[5]Region Västra Göraland, Sahlgrenska University Hospital, Department of Rehabilitation Medicin, Gothenburg, Sweden
[6]Region Västra Götaland, Sahlgrenska University Hospital, Department of Occupational Therapy and Physiotherapy, Gothenburg, Sweden
[7]Department of Clinical Neuroscience, Institute of Neuroscience and Physiology, The Sahlgrenska Academy, University of Gothenburg, Gothenburg, Sweden
[8]Region Västra Götaland, Sahlgrenska University Hospital, Department of Acute Medicine and Geriatrics, Gothenburg, Sweden
[9]Institute of Medicine, Department of Internal Medicine and Clinical Nutrition, Sahlgrenska Academy, University of Gothenburg, Gothenburg, Sweden
[10]Department of Medical Radiation Sciences, Institute of Clinical Sciences, Sahlgrenska Academy, Gothenburg, Sweden
[11]Region Västra Götaland, Sahlgrenska University Hospital, Department of Medical Physics and Biomedical Engineering, Gothenburg, Sweden
[12]Department of Radiology, Institute of Clinical Sciences, Sahlgrenska Academy, Gothenburg, Sweden
[13]Region Västra Götaland, Sahlgrenska University Hospital, Department of Radiology, Gothenburg, Sweden
[14]Region Västra Götaland, Sahlgrenska University Hospital, Department of Neurology, Gothenburg, Sweden

**Acknowledgements** The authors thank research nurses Susanne Nilsson, Ingrid Eriksson, Virginia Villarreal Calvo, Magdalena Hagelberg, and Matilda Errind Arvgård for their excellent work and assistance in recruitment of study participants and for conducting follow-up visits. We also thank the physiotherapists at the stroke unit for performing the motor assessments. Furthermore, we thank the stroke survivors and their kin for generously donating their time to this study. In addition, we thank the Biobank Väst, Bioinformatics and Data Centre, Gothenburg, and SciLifeLab.

**Contributors** CB wrote the manuscript; SK, TMS, KSS, MAM, HS, IB-B, OJ and NDÅ contributed valuable expertise to study outline; KJ, TT and CJ conceived the study and are co-PIs of the study. All authors edited the manuscript and have approved the final version. CJ is the guarantor of the study.

**Funding** This work is supported by the Swedish Research Council (2021–01114), the Swedish state under the agreement between the Swedish government and the county councils, the ALF-agreement (ALFGBG-965328, ALFGBG-965417, ALFGBG-966177 and ALFGBG-942664), the Swedish Heart and Lung Foundation (20220184) and the King Gustaf V:s and Queen Victoria's Foundation.

**Competing interests** None declared.

**Patient and public involvement** Patients and/or the public were involved in the design, or conduct, or reporting, or dissemination plans of this research. Refer to the Methods section for further details.

**Patient consent for publication** Not applicable.

**Provenance and peer review** Not commissioned; externally peer reviewed.

**ORCID iDs**
Cecilia Brännmark http://orcid.org/0000-0003-4848-0480
Tara M Stanne http://orcid.org/0000-0001-9668-0407
Hans Samuelsson http://orcid.org/0000-0003-3753-8317
Margit Alt Murphy http://orcid.org/0000-0002-3192-7787
Katharina S Sunnerhagen http://orcid.org/0000-0002-5940-4400
Oscar Jalnefjord http://orcid.org/0000-0003-2741-5890
Isabella Björkman-Burtscher http://orcid.org/0000-0002-9023-3363
Turgut Tatlisumak http://orcid.org/0000-0002-2430-8988

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
