## [Reviewer comments · BMJ Open]

ARTICLE DETAILS

TITLE (PROVISIONAL)	The FIND Stroke Recovery Study (FIND): Rationale and protocol for a longitudinal observational cohort study of trajectories of recovery and biomarkers post-stroke
AUTHORS	Brännmark, Cecilia; Klasson, Sofia; Stanne, Tara; Samuelsson, Hans; Alt Murphy, Margit; Sunnerhagen, Katharina; Åberg, N. David; Jalnefjord, Oscar; Björkman-Burtscher, Isabella; Jood, Katarina; Tatlisumak, Turgut; Jern, Christina

VERSION 1 – REVIEW

REVIEWER	Zhu, Luwen Heilongjiang University of Chinese Medicine, Fourth Affiliated Hospital
REVIEW RETURNED	10-Mar-2023

GENERAL COMMENTS	Cecilia Brännmark et al. present a protocol for recovery trajectories and biomarkers after stroke. FIND is a clinically valuable longitudinal cohort study with complex phenotypes and repeated assessments of outcome measures across multiple domains (cognitive, neurological) to inform the time course of recovery for patients and healers, combined with precise biomarkers (brain imaging, proteins, RNA, and DNA for multi-omics) to analyze stroke. Our concern was the successful completion of the study with such a large longitudinal study and a long follow-up. In addition, we know that only patients with mild to moderate stroke are better coordinated, and the heterogeneity due to differences in patient severity is undeniable to us.
--

REVIEWER	Demeyere, Nele University of Oxford Medical Sciences Division, Nuffield Department of Clinical Neurosciences
REVIEW RETURNED	13-Mar-2023

GENERAL COMMENTS	The FIND study protocol describes an ongoing longitudinal stroke study, aiming to conclude in 2025. The focus is on understanding recovery trajectories, and a series of measures assessing areas of functional, motor, and cognitive changes after stroke along with blood biomarkers and neuroimaging are conducted at several timepoints post stroke, up to 5 years. I only have comments for further clarifications which would be helpful to be made: • Given start of recruitment in 2018, few people will be able to be followed up 5 years post stroke, unless the follow up will continue
--

	for a further 5 years following the conclusion of recruitment? Please can this be clarified?  • Similarly, given the study is already underway and in fact over half-way, please include an update on numbers so far and anticipated final numbers based on current recruitment Introduction  • Authors note up to 74% cognitive impairment early post stroke, but reference an older neuropsychology paper with extensive batteries including only mild stroke. There are several studies with higher incidence acutely where more inclusive recruitment was used - e.g. Jaillard 2009: 91%, 81% Demeyere et al 2016. I would say it is fair to assume that almost everyone will have some form of cognitive impairment if you assess sensitively (e.g. see some recent ocs-plus work of ours in a rehab setting, so recruiting more severely affected patients – Webb et al 2021) • Re references on cognitive recovery as well as decline : these might also be of interest to the authors : Del Ser et al 2005 and Demeyere et al 2019 Methods:  • Exclusion criteria state: “iii) patients considered unlikely to be able to participate in or to understand and/or comply with study procedures during follow-up visits” -> those with clear cognitive impairment are surely one of the key subgroups of interest in a study looking at cognitive recovery? (see also suggestion re limitations at end) -> please reconsider wording on 'compliance', it is a very paternalistic word. At this point I thought perhaps the exclusion criteria meant to say people who cannot provide informed consent to take part, rather than a view of the researcher or health professional on likely compliance? -> However, later the paper states people who cannot provide consent will be included if next of kin 'consents' – see further section under ethics. • Representatives instead of representants ? • Protocol – covid adaptations: please add additional materials to make clear how protocol was adapted, which measures were used and their validity for remote testing. Will the data be of a mixed type of remote and in-person? How will this be dealt with in analyses? Are they really equivalent ? • Please clarify self-assessed SIS in acute setting ? Is this used at the acute timepoint (which would seem to be invalid for several measures on impact on daily life given not yet returned to home situation ? • “Cognitive function is assessed by The Montreal Cognitive Assessment (MoCA)25. Participants with aphasia also do the 15-Item Boston Naming Test (BNT) items of language comprehension from Addenbrooke's Cognitive Examination-Revised test (ACE-R) and from Norsk grunntest for afasi (NGTA)”
--	--

	It seems then that there is a big focus on verbal memory for the MoCA and that everyone with aphasia only gets a language test, but no tests of any other domains likely to be affected. For a study focussed on domain-specific cognitive recovery, there seems to be missing measures of domain specific cognition, known to be highly prevalent impairments in stroke such as neglect, executive dysfunction, reading/writing, praxis etc. Please make explicit limitation I see from Figure 1 that there are a few more cog measures than stated in above quote. Abbreviations in the figure need making explicit for the reader in the table caption. It seems from this that most cog measures are only starting at 1 year post stroke, so will not inform about acute to 1 year recovery/ cog changes. Please make this explicit (also still not including many key areas of cognition – see above)  • Note on domain specific cognition definition: arguably attention and exec dysfunction are domain general measures which are known to overall impact domain specific functions like language, number, praxis. Most of the additional cog measures seem to reflect more indepth domain general cognition, rather than addressing more stroke specific domains ? • Please clarify and justify the focus on additional measures for those <55 only. Why not for all ? • Primary objective: This seems much too extensive for a primary objective to also look at individual variations in recovery in all measures. This would seem to lead to an explosion of analyses and therefore struggle with power and multiple comparisons. I suggest this is cleaned up, and made more explicit and tied to the primary outcome measures (for which will you have power to do what – if exploratory, needs to move to secondary objectives • Ethics “Written informed consent is obtained from all participants prior to enrolment. For participants who are unable to communicate, consent is obtained from their next-of-kin.” This needs to be clarified: next of kin cannot give consent for someone else - they either give assent, or consultee consent procedures are used. Please check terminology used. As a note: If there is only a communication problem, people may be able to still give informed consent themselves, and make clear they are happy to participate, which can then be observed informed consent. Please clarify the consent procedures more explicitly. How was it determined when people could not communicate? How was this stated in the ethical protocol that was approved? Please also note for your reporting later on you will need to be explicit about how many people could not provide informed consent. Discussion
--	--

	A clear and more comprehensive overview of recent and ongoing studies needs to be included in the justification already in the introduction, not just here in discussion. Please also reference to studies like the young stroke ODYSSEY study (especially given a seeming focus on extra testing in <55s). Also references to the R4VAD study and APPLE may be warranted. R4VAD also has significant imaging, so worth a note on this? to note, i think these are all complementary, and agree re need for more of these studies and aggregating of data and validation of findings. Just would be good to have a more comprehensive intro on what is currently ongoing  • " higher likelihood for withdrawal from the study follow ups in severe stroke Patients" just a note for the authors - not a comment as not published yet, but we actually found we lost similar numbers of severe and very mild stroke in our follow ups, as several of our mildest stroke survivors preferred not to be reminded of their stroke, said they were fine now, had returned to work and lacked time etc. So attrition seems to happen on both ends of the spectrum of severity, keeping the moderate strokes most prominently engaged in long term research Strengths and limitations points at start:  • Authors state heterogeneity at recruitment as a limitation, but this reflects the clinical reality in which any kind of predictive modelling will ultimately be implemented, so not sure this is a true limitation. A more obvious limitation to me would seem not including those with clear cognitive impairment at start, and only taking language follow up measures from those with aphasia instead of more full cognitive functioning across other domains (with aphasia friendly measures).
--	---

VERSION 1 – AUTHOR RESPONSE

We thank you for the thoughtful review comments that have helped to improve our manuscript entitled “The FIND Stroke Recovery Study (FIND): Rationale and protocol for a longitudinal observational cohort study of trajectories of recovery and biomarkers post-stroke”. We were pleased to see that the Reviewers found our study interesting. We have made revisions in the manuscript in accordance with the comments and detailed point-by-point responses are enclosed. All changes in our manuscript have been highlighted in red font.

In light of both reviewer’s comments, we have added some information on our previous experiences of performing longitudinal studies of stroke with follow-ups with neurological and cognitive testing. Given the comments by Reviewer 2, we have modified the Methods section to provide more detailed information about the cognitive tests that are used in this study and which cognitive domains they capture. We have also added a new Table (Table 1) summarizing this information. As suggested by Reviewer 2, we have cited some additional publications, and revised the Introduction to this effect. Finally, we have expanded the Discussion section to meet the comments by Reviewer 2s and cited the suggested additional references. As a result, the word count of the enclosed manuscript (excluding title page, tables, references and abstract) has increased to 4,544 words, and thus now

exceeds your word-limit. We hope that this word count and the number of references are acceptable. If not, please advise, and we will conduct further revisions to comply with necessary requirements.

VERSION 2 – REVIEW

REVIEWER	Demeyere, Nele University of Oxford Medical Sciences Division, Nuffield Department of Clinical Neurosciences
REVIEW RETURNED	19-Apr-2023
GENERAL COMMENTS	Thank you for carefully addressing all my comments and good luck with the study.